# Structure, Activity and Function of the Suv39h1 and Suv39h2 Protein Lysine Methyltransferases

**DOI:** 10.3390/life11070703

**Published:** 2021-07-16

**Authors:** Sara Weirich, Mina S. Khella, Albert Jeltsch

**Affiliations:** 1Institute of Biochemistry and Technical Biochemistry, University of Stuttgart, Allmandring 31, 70569 Stuttgart, Germany; sara.weirich@ibtb.uni-stuttgart.de (S.W.); mina.saad@ibtb.uni-stuttgart.de (M.S.K.); 2Biochemistry Department, Faculty of Pharmacy, Ain Shams University, African Union Organization Street, Abbassia, Cairo 11566, Egypt

**Keywords:** protein lysine methylation, H3K9 methylation, PKMT, enzyme specificity, enzyme regulation, heterochromatin, protein post-translational modification

## Abstract

SUV39H1 and SUV39H2 were the first protein lysine methyltransferases that were identified more than 20 years ago. Both enzymes introduce di- and trimethylation at histone H3 lysine 9 (H3K9) and have important roles in the maintenance of heterochromatin and gene repression. They consist of a catalytically active SET domain and a chromodomain, which binds H3K9me2/3 and has roles in enzyme targeting and regulation. The heterochromatic targeting of SUV39H enzymes is further enhanced by the interaction with HP1 proteins and repeat-associated RNA. SUV39H1 and SUV39H2 recognize an RKST motif with additional residues on both sides, mainly K4 in the case of SUV39H1 and G12 in the case of SUV39H2. Both SUV39H enzymes methylate different non-histone proteins including RAG2, DOT1L, SET8 and HupB in the case of SUV39H1 and LSD1 in the case of SUV39H2. Both enzymes are expressed in embryonic cells and have broad expression profiles in the adult body. SUV39H1 shows little tissue preference except thymus, while SUV39H2 is more highly expressed in the brain, testis and thymus. Both enzymes are connected to cancer, having oncogenic or tumor-suppressive roles depending on the tumor type. In addition, SUV39H2 has roles in the brain during early neurodevelopment.

## 1. Introduction

The unstructured N-terminal tails of the histone proteins protrude from the core nucleosome and contain complex patterns of post-translational modifications (PTMs), including the methylation of lysine and arginine residues, lysine acetylation and the phosphorylation of serine and threonine [1,2,3,4]. These PTMs regulate many features of chromatin biology, gene expression and play a central role in developmental processes of multicellular organisms. In addition, aberrant histone PTMs are implicated in many diseases, such as cancer [5,6]. Acting in concert with DNA methylation and H4K20me3, H3K9me3 is a hallmark of constitutive heterochromatin in eukaryotes [7,8,9,10] and it is also enriched in silenced genes [11]. The suppressor of the variegation 3–9 gene has been genetically identified in screens for suppressors of position effect variegation in D. melanogaster in 1994 [12]. In 2000, its human homolog 1 (SUV39H1, also known as KMT1A) was biochemically identified as the first human protein lysine methyltransferase (PKMT) [13]. It introduces H3K9me3 together with a second human paralog called SUV39H2 (KMT1B) [14], and through H3K9me3 generation both of these enzymes have essential roles in heterochromatin formation and gene silencing. In addition, SUV39H1 and SUV39H2 were shown to methylate different non-histone substrate proteins, with essential functions in regulating protein stability, activity and protein–protein interactions (see below). The SUV39 PKMTs and their function in heterochromatin formation are evolutionarily conserved and orthologous proteins can be detected in most organisms from fission yeast to humans including plants [15,16].

## 2. Domain Architecture and Structure of SUV39 Enzymes

SUV39H1 and SUV39H2 consist of two-conserved domains, one SET- and one chromodomain (Figure 1). The amino acid sequences of SUV39H1 and SUV39H2 are highly conserved, with 56% amino acid identity over the entire protein alignment. The SET (Su(var)3–9, Enhancer-of-zeste, Trithorax) domain is the catalytic domain of one large group of PKMTs, called SET-domain PKMTs [17,18]. The structure of the SUV39H2 SET domain has been solved and it shows a high similarity to the known SET domain structures of other H3K9 PKMTs such as Dim-5 or G9a [19]. This domain binds the methyl group donor S-adenosyl-L-methionine (AdoMet) and brings it in close contact to the target lysine residue in its active site pocket. Chromodomains are methylated lysine binding modules [20,21]. While the SUV39H1 chromodomain was shown to recognize H3K9me2/3 [22], the function of the SUV39H2 chromodomain has not yet been confirmed.

### 2.1. Structure and Biochemical Properties of the SET Domain

One large group of PKMTs contains a SET domain as the catalytically active part, which consists of approximately 130 amino acids [17,18,23]. The SET domain comprises several small β-sheets that surround a knot-like structure in which the C-terminus of the protein is thread through an opening of a short loop in the preceding amino acid sequences. This structure brings together the two most-conserved motifs (NH(S/C)xxPN and ELx(F/Y)DY, where x denotes any amino acid residue) of the SET domain and forms the active site of the enzyme next to the AdoMet binding pocket and substrate peptide binding cleft. It is packed together with a Post-SET, Pre-SET or an additional I-SET domain that is inserted into the core SET domain.

The SET domain of SUV39H2 (Figure 2) has been structurally characterized and shown to contain an additional N-SET region, which is N-terminal to the Pre-SET regions and wraps around the core SET domain [19]. The H3K9 peptide binds in a groove formed by the I-SET and Post-SET domains, where it contacts the enzyme with backbone and side-chain interactions. Thus far, no structure has been solved for the SET domain of SUV39H1 but based on the amino acid sequence similarity, the overall folding and peptide interactions can be expected to be similar.

#### 2.1.1. Biochemical Properties of the SUV39H1 SET Domain

SUV39H1 is able to introduce trimethylation at H3K9 in vitro, but the conversion of H3K9me2 into H3K9me3 is slow [13,24,25]. The SET domain of SUV39H1 introduces methyl groups on the H3 substrate in a non-processive manner [25]. Peptide SPOT array methylation experiments in the context of the H3K9 sequence revealed recognition of H3 residues between K4 and G12 with a highly specific readout of R8 (Figure 3A) [26]. Similar to G9a [27] and SUV39H2 (see below) [28], SUV39H1 shows a high specificity for an arginine at the −1 position (R8) (using K9 as reference position), replacing this R by any other amino acid completely abolished the catalytic activity. Apart from this, residues from the −5 to +3 positions are recognized with variable stringency. At the −5 site (K4), lysine and more weakly arginine were preferred. At the −3 position (T6), the enzyme prefers T, S, A and Y. At the −2 position, SUV39H1 accepts several residues, including polar (N, Q), small (A) and hydrophobic (L, P, W) ones. At the +1 position, the positively charged K and R are equally accepted as the native S10. At the +2 position, SUV39H1 tolerates only small amino acids such as A, G and S, in addition to the native amino acid T11 and at the +3 site, G, K and Q are preferred. In agreement with these findings, the catalytic activity of SUV39H1 has been shown to be influenced by the PTMs of this region of the H3 tail, for example, the trimethylation of K4 has been shown to reduce the activity of SUV39H1 [26,29,30].

#### 2.1.2. Biochemical Properties of the SUV39H2 SET Domain

Similar to SUV39H1, SUV39H2 introduces H3K9me3 in vitro [28,32]. It prefers the unmethylated H3 peptide as substrate [28,33] and the SUV39H2 catalytic SET domain introduces the first two methyl groups into H3K9me0 in a processive reaction [28], but similar to SUV39H1 (see above), the generation of H3K9me3 was slower than the generation of H3K9me2 [28,33]. The recognition of the H3K9 sequence by SUV39H2 has been investigated by peptide array methylation studies, which revealed accurate sequence recognition of the positions R8, S10, T11 and G12. In addition, the residues T6, A7, G13 and K14 were important for the enzyme activity (Figure 3B) [28]. Similar to SUV39H1 (see above) and G9a [27], SUV39H2 critically depends on the recognition of R8. This can be explained on the basis of the SUV39H2-SET domain structure, because D196 in SUV39H2 is ideally positioned to contact R8 with H-bonds. At the C-terminal side of the target lysine, S10 recognition could be mediated by D198 in SUV39H2, which is positioned identically as D209 in Dim-5, which takes over this role in this enzyme [34]. In agreement with the accurate readout of the R8, S10 and T11 positions, the modifications of R8 reduced the methylation activity of SUV39H2 and the phosphorylation of S10 or T11 completely blocked the enzyme [14,28].

Interestingly, the substrate specificity profiles of the two SUV39H enzymes differ from each other (Figure 3). Overall, SUV39H1 has stronger preferences for residues N-terminal to the target lysine, whereas SUV39H2 is more specific for residues C-terminal to the target lysine. One clear difference between SUV39H1 and SUV39H2 is the preference of SUV39H1 for R, K, S and T at the +1 site, where SUV39H2 is more specific and accepts mainly S and, more weakly, T. In general, SUV39H2 is more specific than SUV39H1, because it displays a high preference for the native H3 tail residues at six sequence positions (R8-G12), while SUV39H1 shows stringent readout of only one residue (R8). These differences indicate that the same methylation site on histone H3 is recognized in a different manner and both enzymes could have different non-histone substrate proteins, which may also be one reason explaining the emergence of different SUV39H paralogs in evolution. A similar observation was made for the paralogous SUV4-20H1 and SUV4-20H2 enzymes which also showed overlapping but distinct biological functions and properties [35].

In Clr4, the SUV39H homolog in *S. pombe*, automethylation was observed on K455 and K472, which are located in an autoregulatory loop (ARL) positioned between the SET and post-SET domain [36]. This ARL blocks the active center of the enzyme, but after automethylation, it undergoes a conformational change increasing the enzyme activity [36,37], which potentially connects the intracellular concentration of AdoMet to Clr4 activity [37]. Intriguingly, K392 in SUV39H2, which is analogous to Clr4 K472 and located in a similar flexible loop [19], has been shown to be automethylated as well and accordingly to change the enzyme activity and binding affinity to its substrate proteins [38], suggesting that automethylation might play a role in the regulation of SUV39H2 as well.

### 2.2. Structure and Biochemical Properties of the Chromodomain

Chromodomains are well-known methyllysine interaction domains [20,21]. Structural studies showed that the SUV39H1 chromodomain displays a generally conserved structure compared with other solved chromodomains [22]. The chromodomain fold comprises an N-terminal β-barrel consisting of three anti-parallel strands, which is followed by a long C-terminal α-helix that in the case of the SUV39H1 chromodomain is longer than typically observed with other chromodomains. Biochemical studies documented the specific binding of the SUV39H1 chromodomain to H3K9me3 and, more weakly, H3K9me2, but the overall binding affinities were lower than those observed with other chromodomains [22]. Modelling could identify a trimethyllysine binding cage that is structurally very similar to the one in HP1 proteins (Figure 4).

Biochemical studies revealed that the chromodomain of SUV39H1 inhibits its methyltransferase activity, and this inhibition was relieved by H3K9me3 binding to the chromodomain [39]. Using designer chromatin templates for methylation kinetics, Müller et al. (2016) discovered a two-step activation switch acting in SUV39H1, where H3K9me3 recognition by the chromodomain firstly leads to the anchoring of the enzyme to chromatin. Secondly, the H3K9me3 interaction of the chromodomain led to an allosterically activation of the methylation activity of the SET domain. This process establishes a positive feedback loop for spreading of H3K9me2 and H3K9me3 over extended heterochromatic regions that was shown to be operational in cells as well [39].

In 2017, two additional papers shed more light on the targeting and regulatory role of the SUV39H1 chromodomain [40,41]. Collectively, these papers showed that the chromodomain of SUV39H1 binds to nucleic acids with basic surface residues that are distinct from the trimethyllysine binding cage. Binding was observed to the RNA associated with pericentric heterochromatin, which is retained in cis at its transcription sites. Binding to H3K9me3 and pericentromeric RNA was synergistic and both activities were required for the efficient targeting of SUV39H1 to heterochromatin, H3K9me3 deposition and heterochromatin silencing. The specificity of the nucleic acid binding was partially controversial; while one paper reported binding without sequence preference to ssRNA, ssDNA, dsRNA, dsDNA and RNA/DNA hybrids [40], the second one observed better binding of ssRNA than dsDNA [41]. Regarding the mechanism of the RNA-mediated regulation of SUV39H1, a two-step process similar to that suggested for the H3K9me3-dependent activation of SUV39H1 had been proposed [41]. In this model, the RNA interaction with the chromodomain targets the enzyme and it also leads to an allosteric activation of the catalytic activity of the SET domain by disrupting its inhibitory interaction with the chromodomain. 

The targeting and regulation of SUV39H1 by RNA binding to its chromodomain is also consistent with the finding that the telomeric TERRA RNA associates with this domain and this interaction promotes the accumulation of H3K9me3 at damaged telomeres and end-to-end chromosome fusions [42]. Currently, it is not known if the chromodomain of SUV39H2 has similar roles.

### 2.3. Biochemical Properties of the N-Terminal Part of SUV39H1

The chromodomains of HP1 proteins are critical readers of pericentromeric H3K9me3 [43,44]. Strikingly, SUV39H1 binds directly to HP1 proteins [45] with its N-terminal part [46] and this interaction has been shown to recruit more SUV39H activity to existing H3K9me sites. This process constitutes a self-enforcing feedback loop necessary for the efficient deposition of pericentromeric H3K9me3 [46]. Similarly, in vitro and in vivo data indicated a role of the N-terminal extension to the chromodomain of SUV39H1 in RNA binding [40,41]. In addition, the regulation of SUV39H1 by its N-terminal part and chromodomain has been shown to be under regulation of post-translational modifications in the N-terminal part, because K105 and K123 in the N-terminal part of SUV39H1 were shown to be a target of lysine methylation by SET7/9 in response to DNA damage [47]. The methylation of SUV39H1 reduced its catalytic activity leading to a decreased pericentromeric H3K9me3 and an increased expression of satellite 2 and genome instability [47].

## 3. Biological Roles of SUV39H1 and SUV39H2

### 3.1. Expression Patterns of SUV39H1 and SUV39H2

The expression profiles of SUV39H1 and SUV39H2 in mice are overlapping during embryogenesis, but SUV39H2 remained expressed in adult testis where it is localized at meiotic heterochromatin [14]. In human, SUV39H1 shows little tissue specificity, despite some enrichment in thymus (Figure 5). SUV39H2 is ubiquitously expressed as well, but in adult tissues, the expression is enriched in cerebellum and testis (Figure 5). The expression of SUV39H1 has been observed to decline with age in hematopoietic stem cells [48]. This was shown to lead to a global decrease in H3K9me3 and perturbed heterochromatin function. SUV39H1 was found to be a target of microRNA miR-125b, the expression of which increases with age in human HSC [48]. Moreover, SUV39H1-mediated H3K9 trimethylation regulates the expression of several genes, and the dysregulation of SUV39H1 is observed in different cancers [49,50]. SUV39H2 is overexpressed in many cancer tissues, such as leukemia, lymphomas, lung cancer, breast cancer, colorectal cancer, gastric cancer and hepatocellular cancer [51]. It was found that SUV39H2 is degraded through the ubiquitin-proteasomal pathway and its half-life was reduced by interaction with the translationally controlled tumor protein (TCTP) [52].

### 3.2. Summary of the Functions of H3K9me2/3

As mentioned above, H3K9me3 is a key feature of constitutive heterochromatin in eukaryotes [7,8,9,10] and it also has roles in gene silencing in euchromatic regions [11]. While single SUV39H knock-out mice are viable, the deletion of both SUV39H1 and SUV39H2 is lethal, indicating that the roles of both enzymes are (at least partially) overlapping [13,54]. SUV39H1/2 double knock-out (SUV39H dn) resulted in a drastic loss of pericentric H3K9 trimethylation and also led to chromosomal instabilities [24,54,55]. SUV39H dn cells show severely diminished H3K9me3 levels over the pericentromere, resulting in a lack of accumulation of HP1 proteins and chromosomal instabilities [44,54,55]. In vivo, both SUV39H1 and SUV39H2 introduce H3K9me3 at pericentric heterochromatin, as shown by the finding that the reduction in heterochromatic H3K9 trimethylation in SUV39H dn cells was efficiently recovered by the ectopic expression of either SUV39H1 or SUV39H2 [32,44]. SUV39H1 and SUV39H2 introduced H3K9me3 in the pericentric regions plays a major role in silencing the expression of these regions, thereby repressing ‘selfish’ genetic elements and repetitive DNA and promoting genomic stability [24,54,56].

### 3.3. Chromatin Modification Network of SUV39H1 and SUV39H2

SUV39H1 and SUV39H2 also contribute to the chromatin modification network via different pathways, because HP1 proteins also recruit SUV4-20H enzymes to heterochromatic regions, where they generate H4K20me3 by using the H4K20me1 provided by SET8 as a substrate [57,58,59]. By this mechanism, the H3K9me3 introduced by SUV39H enzymes indirectly stimulates the generation of H4K20me3, another characteristic heterochromatic histone tail modification (Figure 6B). In fact, the knock-out of the SUV39H enzymes has also been shown to lead to decreased levels of heterochromatic H4K20me3 [58].

Another poorly understood observation is that SUV39H1 exists in multimeric complexes with the other H3K9 PKMTs such as G9a, GLP and SETDB1 (Figure 6C) and the deletion of SUV39H1 destabilizes the corresponding proteins and leads to a decrease in the H3K9 methylation signal at the global level [60]. Moreover, in SUV39H or G9a null cells, the remaining H3K9 PKMTs are destabilized at the protein level, indicating that the integrity of these PKMTs is interdependent. In this work, it was also shown that all four H3K9-specific PKMTs are recruited not only to major satellite repeats, a known SUV39H1 genomic target, but also to multiple G9a target genes [60]. Moreover, the functional cooperation between the four H3K9 PKMTs was demonstrated in the regulation of known G9a target genes.

### 3.4. Non-Histone Substrates of SUV39H1 and SUV39H2

As described above, the specificity profile of SUV39H1 differs from SUV39H2, suggesting that these paralogs could have non-redundant functions in the methylation of non-histone proteins. Based on the specificity profile, several SUV39H1 non-histone substrates were identified [26]. The methylation of RAG2, SET8 and DOT1L was confirmed in cells, which all have important roles in chromatin regulation (Figure 6A–C). The SUV39H1-mediated methylation of SET8 was shown to allosterically stimulate its activity [26]. The SET8 PKMT generates monomethylated H4K20 [61,62,63] that is the substrate used by the SUV4-20H enzymes for the generation of H4K20me3 [58,59,64]. This indicates that SUV39H1 controls heterochromatic H4K20 trimethylation through the following two processes: Firstly, SUV39H1-introduced H3K9me3 recruits HP1 proteins that recruit SUV4-20 enzymes, and secondly, it stimulates SET8 to generate more H4K20me1, which is used by SUV4-20H as a substrate (Figure 6B).

Other non-histone substrates of SUV39H1 also have chromatin associated roles: The methylation of RAG2 by SUV39H1 occurs within its NLS and it was shown to alter its sub-nuclear localization [26]. This observation suggests that SUV39H1 could have a direct influence on VDJ recombination catalyzed by RAG2, which is in agreement with data showing that SUV39H1 regulates class switch recombination in B cells [65] and H3K9me3 is associated with this process [66]. This process also contributes to the crosstalk of SUV39H1 with H3K4 methylation because RAG2 is a reader of H3K4me3 (Figure 6C) and H3K4me3 inhibits SUV39H1 activity.

DOT1L is an evolutionarily conserved 7-beta-strand histone PKMT specific for lysine 79 of H3 (H3K79), which has important roles in development and cancer [67]. DOT1L-deficient mouse embryos show reduced levels of heterochromatic H3K9me3 and H4K20me3 marks at centromeres and telomeres indicating that DOT1L plays an important role in heterochromatin formation as well [68]. Conversely, SUV39H1 can also methylate DOT1L, but the biological effects of this methylation event need further investigation [26].

Furthermore, SUV39H1 has been connected to immune function and bacterial infections, because the mycobacterial histone-like HupB protein has been shown to be methylated by SUV39H1 and this process participates in host defense [69]. The SUV39H1 methylation of HupB reduced the survival of mycobacteria inside host cells and it reduced the ability of mycobacteria to form biofilms.

SUV39H2 was found to trimethylate LSD1 at K322 [70] creating a crosstalk of SUV39H2 with H3K4 methylation, because LSD1 has a role in the removal of H3K4me1/2 (Figure 6C). SUV39H2-induced LSD1 methylation suppresses LSD1 polyubiquitination and subsequent degradation, revealing a novel regulatory mechanism of LSD1 in human cancer cells (Figure 6C). SUV39H2 was also reported to methylate K134 of H2AX and stimulate H2AX phosphorylation during DNA damage response [71]. However, the sequence context of H2AX-K134 differs from the specificity of SUV39H2 [28] and in vitro methylation of H2AX could not be confirmed in an independent study [72].

### 3.5. Connections to Diseases

As described above, SUV39H1 and H3K9me3 are predominately associated with the generation and maintenance of constitutive heterochromatin. In mammals, defective pericentric heterochromatin and aberrant transcription of pericentric repeats are associated with genomic instability and cancer [73,74]. These defects in constitutive heterochromatin are most evident in SUV39H1 and SUV39H2 double knockout mice, which exhibit reduced embryonic viability, small stature, chromosome instability, an increased risk of tumor formation and male infertility owing to defective spermatogenesis [54]. Both SUV39H1 and SUV39H2 are prognostic markers for different cancers, but dependent on the tumor type, either a high or low expression of SUV39H1 constitutes a risk, while a high expression of SUV39H2 is unfavorable in most cases (Figure 7).

Human SUV39H1 has been implicated in a variety of complex biological processes such as DNA damage repair [75,76,77], telomere maintenance [42,78], cell differentiation [79,80] and aging [81]. Several lines of evidence documented a tumor suppressive role of SUV39H1 through its stabilization and silencing of heterochromatin. It has been found that SUV39H-deficient mice develop B-cell lymphomas with increased frequencies [54], and SUV39H1 was observed to be downregulated in many leukemias [82]. The protective effect of SUV39H1 in leukemia was validated in mouse models using SUV39H1 overexpression or knockdown and the data provided a direct link between SUV39H1 and AML via the silencing of HOXB13 and SIC1 [82]. Similarly, tumorigenesis driven by Ras or Myc is accelerated by the loss of SUV39H1 [83,84]. Moreover, SUV39H1 was shown to reduce Cyclin D1 expression and, by this, trigger cell cycle arrest [85].

Accumulating evidence indicates that SUV39H2 acts mainly as an oncogene that contributes to the initiation and progression of cancers including invasion and metastasis [51]. As mentioned above, SUV39H2 is overexpressed in many cancers [51], including lung cancer [71,86], acute lymphoblastic leukemia [87], osteosarcoma [88] and glioma [89], and SUV39H2 knockdown resulted in the inhibition of glioma cell growth [89]. The important role of SUV39H2 in cancer is further illustrated by the finding that somatic mutations in this (and other PKMTs) are observed in tumor tissues [6,90]. Moreover, SUV39H2 has been connected with cancer through its regulatory effect on LSD1 [70] and several cancer relevant genes which mainly act as tumor suppressor genes have been shown to be repressed by SUV39H2 overexpression including FAS, P16, P21 and Twist1, while oncogenes like PSA and C-myc are overexpressed [51]. Another study showed that SUV39H2 promotes colorectal cancer proliferation and metastasis via tri-methylation of the SLIT1 promoter and suppression of SLIT1 transcription [91]. In addition, SUV39H2 downregulates the hedgehog interacting protein in glioma cells, thereby promoting hedgehog signaling [89]. Another study identified SUV39H2 as a tumor suppressor and showed that SUV39H2 overexpression in a non-small cell lung cancer (NSCLC) cell line leads to the inhibition of cell growth and proliferation by inducing G_1_ cell cycle arrest [52].

In concordance with its high expression in cerebellum, SUV39H2 has also been connected with neuronal effects. It has been shown that stress-induced H3K9 methylation in the hippocampus was correlated with an upregulation of SUV39H2, suggesting that the enzyme plays a functional role in this process [92]. Recently, the A211S loss of function variant of SUV39H2 has been identified in autism spectrum disorder patients, where it causes altered H3K9-trimethylation and the dysregulation of protocadherin β cluster (Pcdhb cluster) genes in the developing brain [93]. This paper provided direct evidence of the role of SUV39H2 in autism spectrum disorder, and it discovered a molecular pathway of SUV39H2 dysfunction leading to H3K9me3 deficiency, followed by an elevated expression of Pcdhb cluster genes during early neurodevelopment.

Moreover, the SUV39H2 N324K loss of function mutation [28] has been identified to cause hereditary nasal parakeratosis in Labrador Retriever dogs [94], which is a monogenic, inherited, autosomal recessive disorder. Defects in the differentiation of the specialized nasal epidermis cells in affected dogs lead to the formation of crusts and fissures in the nasal planum already in young age, while the animals are otherwise healthy. As differentiation of the nasal epidermis involves selective activation of specific olfactory receptors and silencing of all others, SUV39H2 appears to have a function in this process. This process has been observed to be under the control of the G9a and GLP H3K9 methyltransferases as well as LSD1 [95], which all are connected to SUV39H2, as described above.

## 4. Perspectives and Outlook

The biological function of a PKMT is intimately connected to its substrates and the changes of the substrates’ properties associated with target lysine methylation. For SUV39H1 and SUV39H2, H3K9 is a main substrate. It needs to be studied systematically, how intrinsic and external signals lead to changes in H3K9me3 levels in the heterochromatin and at defined genomic target loci. Moreover, both enzymes also methylate non-histone proteins. The identification of more non-histone proteins and deeper knowledge about their roles in the cellular signaling network will be important for the complete understanding of the biological role of SUV39H1 and SUV39H2. Combined achievements in both directions will help to understand the roles of SUV39H enzymes in diseases better and they may result in novel and specific therapeutic strategies.

## Figures and Tables

**Figure 1 life-11-00703-f001:**
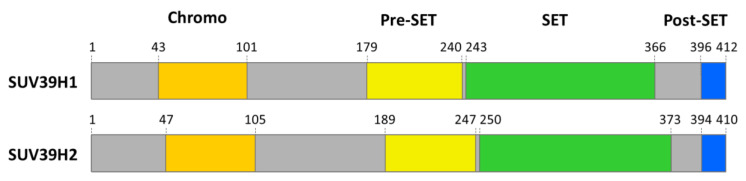
Scheme of the domain structure of human SUV39H1 and SUV39H2. The domain boundaries are indicated as listed in Uniprot.

**Figure 2 life-11-00703-f002:**
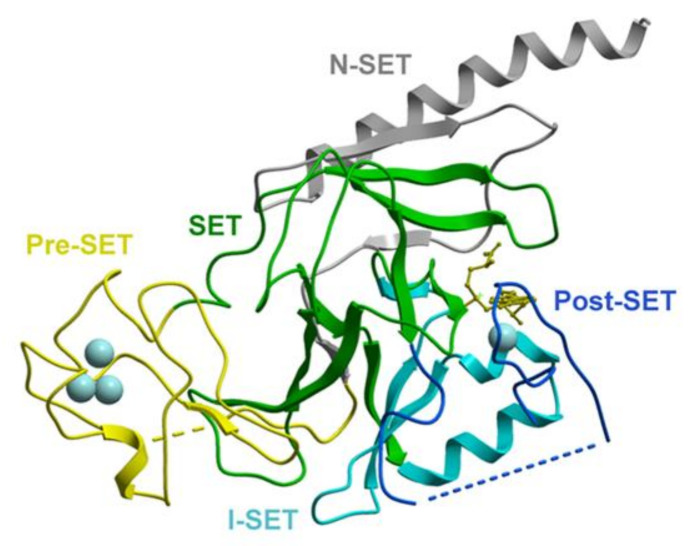
Structure of SUV39H2 in complex with AdoMet. The Pre-Set, SET, I-SET, Post-SET and N-SET domains are highlighted. The co-factor is shown as yellow sticks. Residues flanking un-resolved regions are connected by dotted lines. Taken from [19] with permission.

**Figure 3 life-11-00703-f003:**
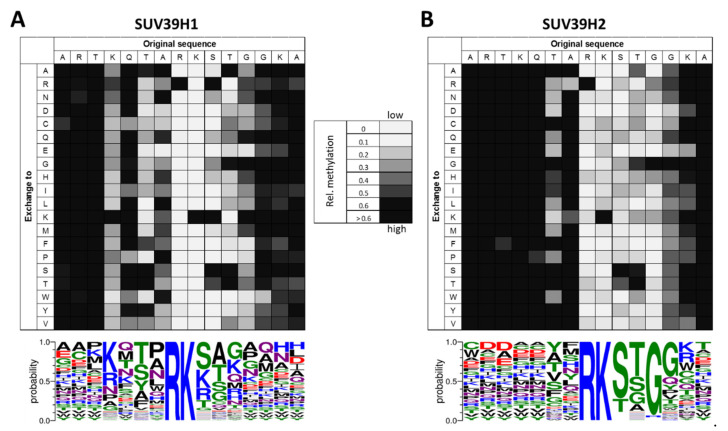
Specificity profiles of SUV39H1 (**A**) and SUV39H2 (**B**). Methylation of peptide substrates containing all possible single amino acid exchanges of the H3 sequence is shown. The horizontal axis represents the sequence of the peptide and in the vertical direction the amino acid that is altered in the corresponding peptide is indicated. Activity is encoded in a grayscale as indicated in the legend. The sequence logo describing the specificity has been prepared with Weblogo3 (http://weblogo.threeplusone.com/ (accessed on 30 April 2021)) [31] and is printed below. Activity data were taken from [26,28].

**Figure 4 life-11-00703-f004:**
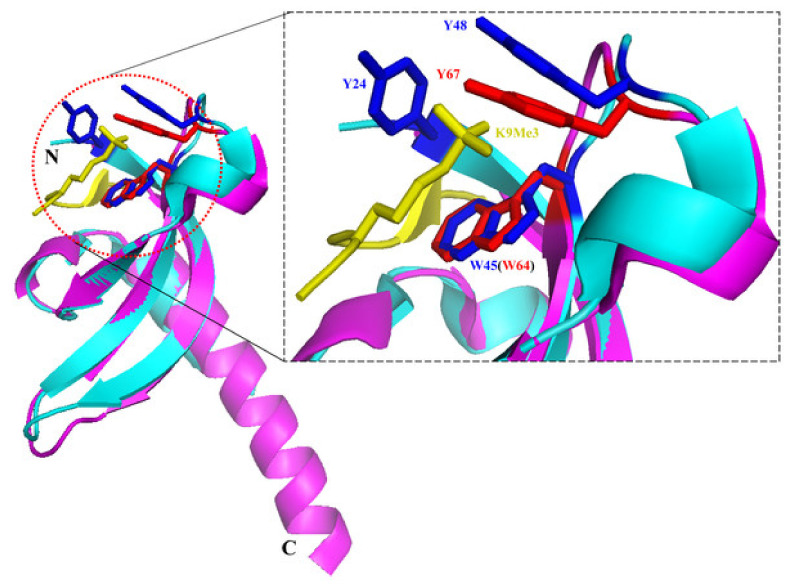
Model of H3K9me3 binding by human SUV39H1 chromodomain. The structures of human SUV39H1 and Drosophila melanogaster HP1 (PDB: 1KNE) chromodomains are aligned and shown in magenta and cyan, respectively [22]. The Y24, W45 and Y48 residues of Drosophila melanogaster HP1 chromodomain that are critical for H3K9me3 binding are shown as sticks in blue. The corresponding residues W64 and Y67 of human SUV39H1 chromodomain are shown as sticks in red. The H3K9me3 peptide present in the Drosophila melanogaster HP1 chromodomain is shown in yellow with trimethylated lysine 9 shown as sticks. Taken from [22] with permission.

**Figure 5 life-11-00703-f005:**
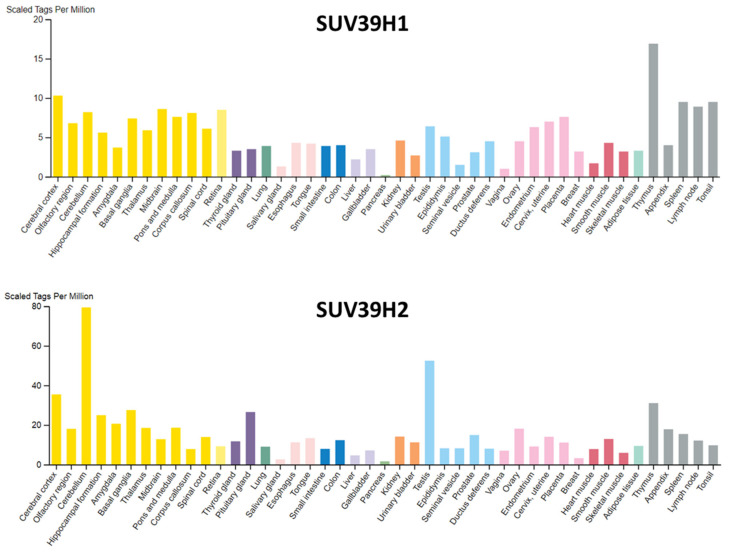
Expression of SUV39H1 and SUV39H2 in human tissues. Data were taken from https://www.proteinatlas.org/ (accessed on 30 April 2021) [53] using the FANTOM5 data set. Data are reported as Scaled Tags per million. Color-coding is based on tissue groups, each consisting of tissues with functional features in common.

**Figure 6 life-11-00703-f006:**
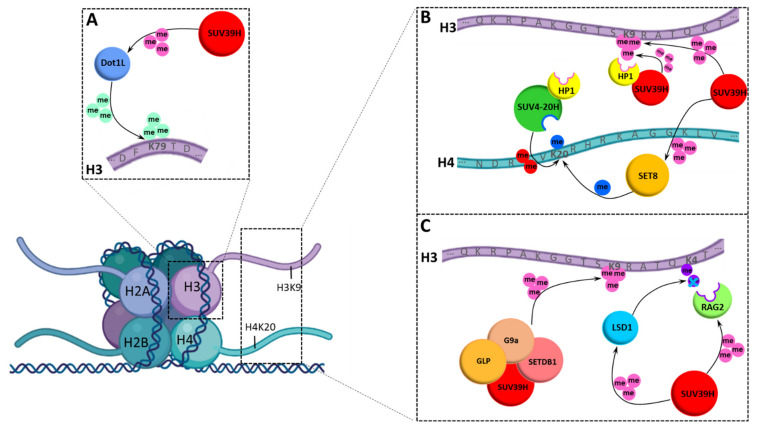
Compilation of the SUV39H centered chromatin network including interacting proteins and complex partners, histone methylation sites and non-histone substrates of SUV39H1 and SUV39H2. The upper left insert (**A**) illustrates SUV39H1 methylation of Dot1L, which itself methylates H3K79. The upper right insert (**B**) shows methylation of H3K9 by SUV39H. This is followed by recruitment of SUV39H and SUV4-20H by HP1 proteins to the H3K9me2/3 sites, leading to the spreading of H3K9me3 and introduction of H4K20me2/3 at H4K20me1 sites. Generation of H4K20me1 by SET8 is stimulated by SUV39H1 mediated methylation. The lower right insert (**C**) features the role of SUV39H in H3K9 methylation as members of the complex of SUV39H with G9a, GPL and SETDB1 PKMTs. SUV39H methylation of the LSD1 (SUV39H2) and RAG2 (SUV39H1) non-histone substrates is shown, which are creating a crosstalk with H3K4 methylation. LSD1 has a role in the removal of H3K4me1/2 and RAG2 is a reader of H3K4me3.

**Figure 7 life-11-00703-f007:**
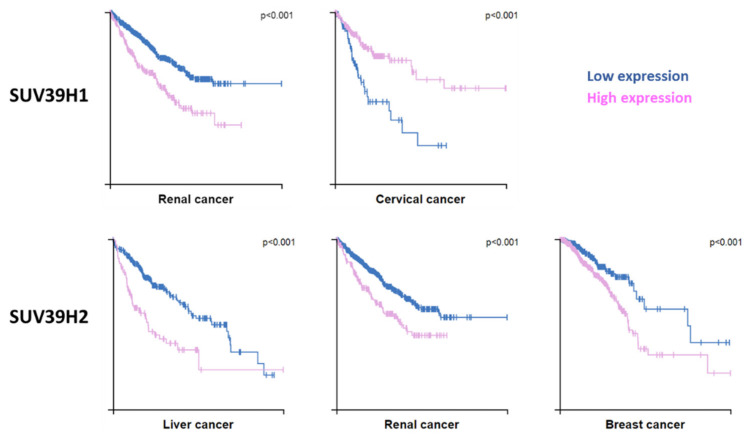
Cancer connection of SUV39H1 and SUV39H2. Kaplan–Meier plots are shown for cancers where high expression of SUV39H1 or SUV39H2 has significant (*p* < 0.001) association with patient survival. Data were retrieved at https://www.proteinatlas.org (accessed on 30 April 2021) [53].

## Data Availability

Not applicable.

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
