# Peer review of "Structure, Activity and Function of the Suv39h1 and Suv39h2 Protein Lysine Methyltransferases"

_life, 2021, doi:10.3390/life11070703_

Round 1
Reviewer 1 Report
The manuscript by Weirich et al. summarizes the structure, activity and function of two lysine methyltransferase enzymes (KMTs) SUV39H1 and SUV39H2. In general, the manuscript is well written and gives a comprehensive overview of these two enzymes. However, I feel the scope of the review is quite limited, as there are many other KMTs known that catalyze the (tri)methylation of the H3K9 site. Therefore, I think the scientific contribution would be greatly increased if the authors would review a larger scope of KMTs that act on this site and go into their (bio)chemical roles in human health and disease. The manuscript by Maas et al. (Int. J. Mol. Sci. 2020, 21(24), 9451) describes these two enzymes in their relation to the methylation of H3K9 already, within the larger area of lysine trimethylation in general. In this regard, a review on a more complete panel of the KMTs that act here, could give a great overview of this class of enzymes and in what processes they play a role. All in all, there is great potential in the work presented here, but only if the authors would expand the scope of their review greatly, giving a more complete view on the very important class of enzymes presented here.
Author Response
Reply: Thank you very much for this positive assessment. We agree that a larger review covering all H3K9 PKMTs would be an interesting topic. However, it has been decided that the scope of this special issue will be on individual enzymes. We believe that this also has advantages, as it provides readers a focused information.
Reviewer 2 Report
The manuscript provides a detailed and thorough summary of the current knowledge on the function of SUV39H1 and SUV39H2 proteins. Their structural and functional differences and similarities are introduced in a meticulous way, encorporating the most recent results in the description.
The text is logical and easy to follow, there are only minor grammatical mistakes.
The manuscript will be helpful for a wide variety of readers interested in histone methylation.
- The subchapter 2.1.1. is duplicated.
- The description of the sequence preferences is a bit too detailed, it would be enough just to mention the key points.
- Figure 6. is a little crowded and difficult to read, as most of the labels are too small. The colour coding seems to carry meaning, but it is not explained in the figure legend. Also, SUV39H and SUV39H1 are both used in the legend and on the figure, with no clear indication of SUV39H2.
- A table summarizing the main similarities/differences of the two enzymes would be a great addition.
Author Response
Reply: Thank you for this positive assessment.
“1. The subchapter 2.1.1. is duplicated.”
Corrected.
“2. The description of the sequence preferences is a bit too detailed, it would be enough just to mention the key points.”
Reply: This description has been shortened.
“3. Figure 6. is a little crowded and difficult to read, as most of the labels are too small. The colour coding seems to carry meaning, but it is not explained in the figure legend. Also, SUV39H and SUV39H1 are both used in the legend and on the figure, with no clear indication of SUV39H2.”
Reply: Thanks a lot. The legend has been rewritten and now everything shown in the figure is much better explained. We enlarged the size of text in the figure where possible. We have checked the resolution of the figure in the original submission and confirmed that it will be readable online.
“4. A table summarizing the main similarities/differences of the two enzymes would be a great addition.”
Reply: We believe squeezing the detailed information into a table may create confusion and propose not to include a table.
Reviewer 3 Report
In the literature, a good review on the structure, activity and function of the SUV39H proteins is missing, then the topic of this manuscript is interesting.
However, a more personal point of view would be appreciated. There are missing good transitions between paragraphs and some are not well structured leading to a difficult understanding. We don’t always clearly see what is important and why.
It is necessary to remove the duplicated paragraph, remove and/or change sentences copied from others publications (see below).
Figure 2, 3 and 4 are taken from others publications. “With permission” seems to be missing in Figure 3.
In details:
Abstract:
“which binds H3K9me3 and has roles in targeting and regulation”: targeting to what? Regulation of what?
It is an enumeration of different characteristics. Connections between them would be appreciate.
Introduction.
It could be better organized and the reading is not easy. The importance of the Suv39h enzymes is not enough emphases. What is their function more precisely? What is the function of Suv39h enzymes in silencing gene expression?
- Domain architecture and structure of SUV39 enzymes
The review starts with “Domain architecture and structure of SUV39 enzymes”. Why is that important? There is missing transition between the introduction and this part.
The targeting mechanisms of Suv39h enzymes to their genomic sequence is not clear.
On Figure 1 there is mention of pre- and post-SET. What is the role of these domains? Not clear.
2.1. Structure and biochemical properties of the SET domain
I found the same sentence in a previous publication (https://www.ncbi.nlm.nih.gov/pmc/articles/PMC2733851/)
“The SET domain folds into several small β-sheets that surround a knot-like structure by threading the C-terminus of the protein through an opening of a short loop formed by a preceding stretch of the protein sequence.”
“This knot-like structure brings together the two most-conserved motifs (NH(S/C)xxPN and ELx(F/Y)DY, where x denotes any amino acid residue) of the SET domain to form an active site immediately next to the methyl-donor-binding pocket and peptide-binding cleft.”
Again:
“N-SET region located N-terminal to the Pre-SET, that wraps around the core SET domain” is from https://europepmc.org/backend/ptpmcrender.fcgi?accid=PMC2797608&blobtype=pdf.
2.1.1. Biochemical properties of the SUV39H1 SET domain
This paragraph is duplicated pages 2 and 3!
3.1 and 3.2 are superficial.
3.2. Functions of H3K9 methylation
What are the differences between Suv39h proteins and the others H3K9 HKMTs? Mono, di and tri-methylation of H3K9?
There is no mention about the repression of gene transcription in euchromatic regions.
Figure 6. Numbers in legend should appear in the figure for clarity.
3.5. Connections to diseases
Well written and interesting part!
- Perspectives and outlook
Interesting! Authors should expand this section.
Author Response
Reply: Thank you for the helpful comments. We have improved writing following your proposals. The numbering of the paragraph under 2.1. has been corrected. Figure 3 was not taken from a paper but it was prepared newly using the published data, as indicted in the legend.
“which binds H3K9me3 and has roles in targeting and regulation”: targeting to what? Regulation of what?
It is an enumeration of different characteristics. Connections between them would be appreciate.”
Reply: The text has been changed to “enzyme targeting and regulation”
“Introduction.
It could be better organized and the reading is not easy. The importance of the Suv39h enzymes is not enough emphases. What is their function more precisely? What is the function of Suv39h enzymes in silencing gene expression?”
Reply: Please note the introduction clearly states “H3K9me3 is a hallmark of constitutive heterochromatin in eukaryotes” and “It introduces H3K9me3”. We have now added “and trough this these both enzymes have essential roles in heterochromatin formation and gene silencing” to clarify this point further. More details are provided in chapter 3.2.
“2 Domain architecture and structure of SUV39 enzymes
The review starts with “Domain architecture and structure of SUV39 enzymes”. Why is that important? There is missing transition between the introduction and this part.”
Reply: We believe it is obvious that a review on a certain enzyme or protein must consider the amino acid sequence and domain composition. These is no further explanation necessary.
“The targeting mechanisms of Suv39h enzymes to their genomic sequence is not clear.”
Reply: Targeting by HP1 is described in the chapter 2.3 as HP1 interaction is mediated by the N-terminal part. This chapter has been expanded to further clarify the concept. Targeting via the Chromodomain is described in the chapter 2.2 in detail.
“On Figure 1 there is mention of pre- and post-SET. What is the role of these domains? Not clear.”
Reply: This is explained in chapter 2.1 where the structure of the SET domain is described “It is packed together with a Post-SET, Pre-SET, or an additional domain (I-SET) inserted into SET domain”.
“2.1. Structure and biochemical properties of the SET domain
I found the same sentence in a previous publication (https://www.ncbi.nlm.nih.gov/pmc/articles/PMC2733851/)
“The SET domain folds into several small β-sheets that surround a knot-like structure by threading the C-terminus of the protein through an opening of a short loop formed by a preceding stretch of the protein sequence.”
“This knot-like structure brings together the two most-conserved motifs (NH(S/C)xxPN and ELx(F/Y)DY, where x denotes any amino acid residue) of the SET domain to form an active site immediately next to the methyl-donor-binding pocket and peptide-binding cleft.”
Again:
“N-SET region located N-terminal to the Pre-SET, that wraps around the core SET domain” is from https://europepmc.org/backend/ptpmcrender.fcgi?accid=PMC2797608&blobtype=pdf.”
Reply: The references were indicated and the sentences just describe a publicly available information. Still we have reformulated the sentences.
“2.1.1. Biochemical properties of the SUV39H1 SET domain
This paragraph is duplicated pages 2 and 3!”
Reply: This mistake has been corrected.
“3.1 and 3.2 are superficial.”
Reply: We respectfully disagree. Information about expression patters is very relevant for readers and a summary of the key biological role (H3K9 methylation) was even requested by this reviewer.
“3.2. Functions of H3K9 methylation
What are the differences between Suv39h proteins and the others H3K9 HKMTs? Mono, di and tri-methylation of H3K9?
There is no mention about the repression of gene transcription in euchromatic regions.”
Reply: The discussion of other H3K9 PKMTs is not within the scope of this review. Gene silencing in euchromatic regions has now been mentioned as well. A more detailed discussion of the functions of H3K9 methylation is beyond the scope of this paper focussing on the SUV39H enzymes. In response to this comment the heading of the 3.2 was changed to "Summary of the funcitons of H3K9me2/3"
"Figure 6. Numbers in legend should appear in the figure for clarity."
Reply: The legend has been rewritten. The inserts are now clearly designated and numbers were inserted in the figure.
“3.5. Connections to diseases
Well written and interesting part!”
Reply: Thank you.
„4 Perspectives and outlook
Interesting! Authors should expand this section.”
Reply: Thank you. We have included our thoughts here and could not find additional points to be mentioned.
Round 2
Reviewer 3 Report
Authors responded to most of my comments and I support publication. However I think that moderate English changes are still required